# NCOR1 Sustains Colorectal Cancer Cell Growth and Protects against Cellular Senescence

**DOI:** 10.3390/cancers13174414

**Published:** 2021-09-01

**Authors:** Stéphanie St-Jean, Ariane Cristina De Castro, Mia Lecours, Christine Jones, Nathalie Rivard, Francis Rodier, Nathalie Perreault, François Boudreau

**Affiliations:** 1Department of Immunology and Cell Biology, Université de Sherbrooke, 3201 Rue Jean-Mignault, Sherbrooke, QC J1E 4K8, Canada; Stephanie.St-Jean2@USherbrooke.ca (S.S.-J.); Ariane.Cristina.De.Castro@USherbrooke.ca (A.C.D.C.); Mia.Lecours@USherbrooke.ca (M.L.); Christine.Jones@USherbrooke.ca (C.J.); Nathalie.Rivard@USherbrooke.ca (N.R.); Nathalie.Perreault@USherbrooke.ca (N.P.); 2Département de Radiologie, Radio-Oncologie et Médecine Nucléaire, Université de Montréal, C.P. 6128, Succursale Centre-Ville, Montreal, QC H3C 3J7, Canada; rodierf@mac.com

**Keywords:** NCOR1, intestinal tumorigenesis, Caco-2/15, HT-29, senescence, *Apc^Min/+^*

## Abstract

**Simple Summary:**

NCOR1 is a scaffold protein that interacts with multiple partners to repress gene transcription. NCOR1 controls immunometabolic functions in several tissues and has been recently shown to protect against experimental colitis in mice. Our laboratory has observed a pro-proliferative role of NCOR1 in normal intestinal epithelial cells. However, it is unclear whether NCOR1 is functionally involved in colon cancer. This study demonstrated that NCOR1 is required for colorectal cancer cell growth. Depletion of NCOR1 caused these cells to become senescent. Transcriptomic signatures confirmed these observations but also predicted the potential for these cells to become pro-invasive. Thus, NCOR1 plays a novel role in preventing cancer-associated senescence and could represent a target for controlling colon cancer progression.

**Abstract:**

NCOR1 is a corepressor that mediates transcriptional repression through its association with nuclear receptors and specific transcription factors. Some evidence supports a role for NCOR1 in neonatal intestinal epithelium maturation and the maintenance of epithelial integrity during experimental colitis in mice. We hypothesized that NCOR1 could control colorectal cancer cell proliferation and tumorigenicity. Conditional intestinal epithelial deletion of *Ncor1* in *Apc^Min/+^* mice resulted in a significant reduction in polyposis. RNAi targeting of NCOR1 in Caco-2/15 and HT-29 cell lines led to a reduction in cell growth, characterized by cellular senescence associated with a secretory phenotype. Tumor growth of HT-29 cells was reduced in the absence of NCOR1 in the mouse xenografts. RNA-seq transcriptome profiling of colon cancer cells confirmed the senescence phenotype in the absence of NCOR1 and predicted the occurrence of a pro-migration cellular signature in this context. SOX2, a transcription factor essential for pluripotency of embryonic stem cells, was induced under these conditions. In conclusion, depletion of NCOR1 reduced intestinal polyposis in mice and caused growth arrest, leading to senescence in human colorectal cell lines. The acquisition of a pro-metastasis signature in the absence of NCOR1 could indicate long-term potential adverse consequences of colon-cancer-induced senescence.

## 1. Introduction

NCOR1 (nuclear corepressor 1) was originally discovered as a protein interactor of thyroid hormone receptors (TR) and retinoic acid receptors (RAR), which repressed the transcription of genes harboring these DNA elements [1,2]. It was further discovered that the transcriptional repressive function of NCOR1 extended beyond interactions with nuclear receptors. For example, NCOR1 can interact with c-Jun and be recruited to AP1 sites on target gene promoters [3]. NCOR1 forms large protein complexes that contain histone deacetylases (HDACs), for which HDAC3 requires a physical interaction with NCOR1 to sustain its catalytic activity [4,5]. Additional proteins of the complex are involved in sensing signals to mediate chromatin corepressor exchange via the action of specific kinases and ubiquitylases [6]. Thus, NCOR1 functions as a large protein platform complex that integrates multiple regulatory proteins involved in epigenetic modifications, leading to chromatin compaction and gene repression.

*Ncor1* deletion in mice has revealed essential functions in embryonic development with lethality at embryonic day 15.5 because of anemia and defects in the terminal erythropoiesis process [7]. To circumvent lethality, a conditional *Ncor1*^−/−^ mouse model was generated [8]. This strategy revealed important roles for NCOR1 in restricting muscle mass and oxidative function [8], reducing obesity through gene transrepression via PPARγ phosphorylation [9], and mediating pro- or anti-inflammatory signatures depending on the tissue or cellular source for which it was depleted [10,11,12]. The conditional deletion of *Ncor1* in intestinal epithelial cells accelerated intestinal maturation in neonatal mice [13], whereas very recent observations with the same deletion model indicated that proliferation of intestinal crypt cells was severely impaired after dextran sulfate sodium (DSS)-induced experimental colitis [14]. Our laboratory has also shown that NCOR1 is crucial for maintaining the proliferation of a normal intestinal epithelial cell line in culture [15]. Collectively, these observations support the important role of NCOR1 as a pro-proliferative entity in intestinal epithelial cells and highlight the possibility that disruption of its function could be beneficial during tumorigenesis.

Colorectal cancer (CRC) is one of the most common cancers worldwide, ranking third in incidence and second in terms of mortality [16]. Significant progress has been made in identifying specific genes associated with predisposition to CRC [17]. Additionally, the disruption of several signaling pathways, such as WNT, JAK-STAT, TGFβ/SMAD, and NOTCH, were also identified as intimately linked with defects in differentiation, angiogenesis, apoptosis, and cell proliferation during CRC [18]. One of the common features of CRC is the ability of cancer cells to lose control of cell cycle checkpoints, leading to abnormal continuous growth. One endogenous mechanism that can inhibit the growth of intestinal epithelial transformed cells is senescence [19]. Cellular senescence is characterized by the complete cessation of cell proliferation associated with cellular aging. The hayflick limit was first described 60 years ago as the maximum number of divisions that cells can make in culture [20], and this was later attributed to telomere shortening [21]. This process can trigger senescence, but additional stressful insults, such as oxidative stress, loss of tumor suppressors, or activation of oncogenes, can initiate senescence [22]. In addition to cell cycle arrest and macromolecular damage, secretion of several factors, including pro-inflammatory molecules, growth factors, and matrix metalloproteinases, occurs in senescent cells, which is referred to as the senescence-associated secretory phenotype (SASP) [23]. Although senescence is primarily regarded as an anti-cancer mechanism, there is now evidence that SASP promotes a pro-inflammatory milieu and, paradoxically, leads to tumor progression [24].

Here, we investigated the role of NCOR1 in intestinal tumorigenesis. Reduction of NCOR1 expression in human CRC cell lines led to drastic growth inhibition, which was associated with senescence. These observations identified a novel biological role for NCOR1 in intestinal neoplasia.

## 2. Materials and Methods

### 2.1. Animals

12.4KbVilCre transgenic mice [25] and *Ncor1^fx/fx^* [8] were used to produce *Apc^Min/+^; Ncor1*^+/+^(control), *Apc^Min/+^*; 12.4KbVilCre/*Ncor1^fx/+^* (heterozygotes; *Ncor1^DIEC/+^*), and *Apc^Min/+^*; 12.4KbVilCre/*Ncor1^fx/fx^* (*Ncor1^DIEC^*) mice on a pure C57BL/6J background. Polyp counts were determined as previously described [26] in accordance with the Institutional Animal Research Review Committee of the University of Sherbrooke (approval ID number 342-17). CD1 nu/nu mice were purchased from Charles River Laboratory and xenograft experiments were approved by the Institutional Animal Research Review Committee of the University of Sherbrooke (approval ID number 276-15B).

### 2.2. Cell Culture

The human embryonic kidney cell line 293T and CRC cell line HT-29 were obtained from the American Type Culture Collection (ATCC). The Caco-2/15 CRC cell line was a gift from Dr. J.-F. Beaulieu (Université de Sherbrooke). 293T and Caco-2/15 cells were cultured in DMEM medium (Wisent, Quebec, QC, Canada), while HT-29 cells were cultured in McCoy’s 5a medium (Wisent). All media were supplemented with 10% fetal bovine serum (Wisent), 2 mM GlutaMAX (Thermo Fisher Scientific, Waltham, MA, USA), 0.01 M HEPES (Wisent), and 100 μg/mL of penicillin/streptomycin (Thermo Fisher Scientific). Cells were cultured at 37 °C with 5% CO_2_. Trypan blue was used to assess viability of adherent and nonadherent cells. Cells were counted with the Countess Automated Cell Counter (Invitrogen, Thermo Fisher Scientific) to assess proliferation.

### 2.3. NCOR1 Knockdown with shRNAs

Subconfluent Caco-2/15 and HT-29 cells were plated in six-well plates. After overnight incubation, cells were infected with lentiviruses containing shRNAs against NCOR1 (Sigma Mission shRNA: shNCOR1_654 clone ID TRCN0000060654; shNCOR1_655 clone ID TRCN0000060655; shNCOR1_656 clone ID TRCN0000060656; shNCOR1_657 clone ID TRCN0000060657). A nontargeting shRNA (shNonTarget Sigma clone ID SHC002) was also used as a control. Freshly established cell lines were generated for each experiment since shNCOR1 constructs led to rapid inhibition of cell growth.

### 2.4. RNA Isolation and qRT-PCR

Total RNA was isolated using the RNeasy Kit (QIAGEN, Mississauga, ON, Canada) and cDNA was synthesized using AMV-RT (Roche, Laval, QC, Canada). Quantitative real-time PCR (RT-qPCR) reactions were prepared using 2 µL of diluted cDNA and FastStart Essential DNA Green Master (Roche). qRT-PCR was performed using a LightCycler 96 system (Roche). Gene expression was normalized relative to the mRNA expression of TATA box-binding protein (*TBP*). Oligonucleotide sequences are available upon request.

### 2.5. Cell Adhesion and SAβ-Galactosidase Assays

Adhesion tests were performed in 6-well plates (Life Sciences, BD Falcon, Tewksbury, MA, USA). Cells (1 × 10^6^) were inoculated into wells and incubated at 37 °C in an environment containing 5% CO_2_ for 16 h. Subsequently, cells in suspension were collected in a 15 mL conical tube, centrifuged at 1000 rpm for 5 min at room temperature, resuspended in a specific volume of culture medium, and counted using the Countess Automated Cell Counter (Invitrogen, Thermo Fisher Scientific). The adhered cells were trypsinized and counted. Senescence-associated (SA) β-galactosidase activity was detected as described elsewhere [27]. Briefly, senescence tests were carried out 7 d after puromycin selection of lentiviral transduced cells. Adhered cells were rinsed twice in PBS 1X and fixed in 2% formaldehyde-0.2% glutaraldehyde for 5 min at room temperature. After two washes in PBS 1X, cells were incubated in freshly prepared staining solution (40 mM citric acid/sodium buffer phosphate, 5 mM K_4_ [Fe (CN)_6_], 5 mM K_3_ [Fe (CN)_6_], 150 mM NaCl, 2 mM MgCl_2_, 1 mg/mL X-Gal, pH 6.0) at 37 °C for 3 h (Caco-2/15) or 8 h (HT-29). Cells were washed twice in PBS 1X and then methanol before letting them air dry. The plates were photographed using a DMIL microscope equipped with a Leica DC300 camera (Leica Camera, Allendale, NJ, USA).

### 2.6. Soft Agarose and Colony-Forming Cell Assays

Soft agarose assays were done as previously described [28]. Briefly, DMEM-2X without phenol-red (Wisent) was supplemented with 20% FBS and mixed 1:1 with melted sterile 1.4% agarose type VII (Sigma Aldrich, Oakville, ON, Canada). Each well of six-well plates was coated with 1 mL of a DMEM-agarose mixture. A total of 35,000 HT-29 shNonTarget or shNCOR1_655 puromycin-selected living cells were added to 7 mL of DMEM-agarose and 2 mL was seeded per well for a total of three wells per assay. Once the agarose solidified, the plates were incubated at 37 °C and 5% CO_2_. DMEM supplemented with 10% FBS without phenol-red was added to the surface of agarose every 2 d. After 1–3 weeks, colonies were stained for 3 h with 1 mL of DMEM containing 0.5 mg/mL MTT (37 °C, 5% CO_2_). Colonies were counted using ImageJ software. Colony-forming assays were performed as previously described [29]. Briefly, 10,000 living cells were seeded in 10 mL of cell culture to obtain a cell density equivalent to 1000 cells/mL per well in 6-well plates. Cells were cultured for up to 3 weeks and the culture medium was changed every 2–3 d. The cell culture medium was removed and the cells were washed twice with 1X PBS. Cells were then stained for 30 min at room temperature using a crystal violet solution (6.0% (*v*/*v*) glutaraldehyde, 0.5% crystal violet (*w*/*v*) diluted in water). The cells were washed twice with H_2_O, dried, and photographed. This experiment was repeated a minimum of three times for each cell line.

### 2.7. Xenografts into Nude Mice

HT-29 cells were transduced with either shNonTarget or shNCOR1_655 lentiviruses and selected with 10 µg/mL of puromycin for 7 d. A total of 1 × 10^6^ living cells per lentivirus condition were suspended in 0.1 mL DMEM. Each suspension was injected into the dorsal subcutaneous tissue of each mouse at four independent sites. Five female nude mice per lentiviral condition were used. Tumor volume was determined by external measurement (d^2^ × D)/2, as previously described [28].

### 2.8. Western Blot and ELISA

Adherent and nonadherent cells were lysed at 4 °C with stirring for 30 min in RIPA 1× (50 mM Tris, pH 7.4, 150 mM NaCl, 1% Nonidet P-40, 0.5% Triton X-100, 1 mM EDTA, 0.2% SDS, and 0.5% sodium deoxycholate), to which protease inhibitors were added immediately before use (1% protease inhibitor cocktail (Sigma Aldrich), 0.2 mM Na_3_PO_4_ (Sigma Aldrich), and 50 mM NaF (Sigma Aldrich)). The lysates were sonicated using a Sonic Dismembrator Model 120 (Thermo Fisher Scientific). BCA protein assays were performed to quantify the proteins. Protein extracts (20–40 µg) were loaded and migrated on an SDS-PAGE NuPAGE gel (Invitrogen, Thermo Fisher Scientific). Proteins were then transferred onto a PVDF membrane (Roche), which was blocked in 10% non-fat milk diluted in PBS/0.1% Tween20. To monitor the expression of NCOR1, PARP, Histone H2A.X phosphorylation of Ser-139, SOX2, and β-actin in cell extracts, the following antibodies were used: anti-NCOR1 antibody (Abcam Toronto, ON, Canada; ab3482), anti-PARP antibody (Cell Signaling Technology, Danvers, MA, USA; 9542), anti-phospho-Histone H2A.X (Ser139) (Cell Signaling Technology; 9718), anti-SOX2 (Cell Signaling Technology; 3579), and anti-actin antibody clone C4 (MilliporeSigma; MAB1501R). HRP-linked secondary antibodies were used in combination with ECL-Prime Western Blotting Detection Reagent (GE Healthcare, Mississauga, ON, Canada) to detect signals. Cytokine detection in cell culture conditioned media was performed using the Proteome Profiler Human Cytokine Array Kit (R&D Systems, Minneapolis, MN, USA), CXCL1/GRO Alpha Quantikine ELISA Kit (R&D Systems; DGR00), Migration Inhibitory Factor (MIF) Quantikine ELISA Kit (R&D Systems; DMF00B), CXCL8/IL-8 Quantikine ELISA Kit (R&D Systems; D8000C), and ICAM-1/CD54 Quantikine ELISA Kit (R&D Systems; DCD540), according to the manufacturer’s instructions. Original autoradiograms are shown in Appendix A.

### 2.9. RNA-Seq and Bioinformatics

Caco-2/15 and HT-29 cells were transduced in triplicate with either shNonTarget or shNCOR1_655 lentiviruses and selected with 10 µg/mL of puromycin for 7 d. Total RNA from adhered cells was isolated as described above and quantified using a NanoDrop Spectrophotometer ND-1000 (NanoDrop Technologies, Thermo Fisher Scientific), and RNA integrity was assessed using a 2100 Bioanalyzer (Agilent Technologies, St-Laurent, QC, Canada) instrument. Libraries were generated from 250 ng of total RNA as follows. mRNA enrichment was performed using the NEBNext Poly(A) Magnetic Isolation Module (New England BioLabs, Whitby, ON, Canada). cDNA synthesis was achieved using NEBNext RNA First Strand Synthesis and NEBNext Ultra Directional RNA Second Strand Synthesis Module (New England BioLabs). The remaining library preparation steps were performed using the NEBNext Ultra II DNA Library Prep Kit for Illumina (New England BioLabs). Adapters and PCR primers were purchased from New England BioLabs. Libraries were quantified using the Quant-iT™ PicoGreen^®^ dsDNA Assay Kit (Life Technologies, Thermo Fisher Scientific) and the Kapa Illumina GA with the Revised Primers-SYBR Fast Universal Kit (Kapa Biosystems, MilliporeSigma). The average fragment sizes were determined using a LabChip GX instrument (PerkinElmer). The libraries were normalized, denatured in 0.05 N NaOH, diluted to 200 pM, and neutralized using the HT1 buffer. Clustering was performed on an Illumina cBot (Illumina, Vancouver, BC, Canada) and the flow cell was run on a HiSeq 2000 PE100, following the manufacturer’s instructions. The bcl2fastq v2.20 program was then used to demultiplex samples and generate fastq reads. Bioinformatics analysis of differential gene expression obtained from RNA sequencing was performed by the Canadian Center for Computational Genomics-Montreal Node (https://www.computationalgenomics.ca/montreal-node/ (31 August 2021)). Analysis of canonical pathway enrichment was performed using the Ingenuity Pathway Analysis (IPA) software (QIAGEN). Only pathways with a *p*-value < 0.05 were considered significant.

### 2.10. Statistics

All data are expressed as the mean ± SEM. Groups were compared using a two-way analysis of variance (ANOVA) or Student’s *t*-test unless otherwise specified in the figure legend (GraphPad Prism 8, GraphPad Software, San Diego, CA, USA). Statistical significance was defined as *p* < 0.05.

## 3. Results

### 3.1. Loss of Intestinal Epithelial NCOR1 Reduces Polyposis in Apc^Min/+^ Mice

To determine whether reducing NCOR1 expression could impact intestinal tumorigenesis, we generated mice lacking intestinal epithelial NCOR1 as previously described [8], but here into a C57BL/6J *Apc^Min/+^* background. The conditional deletion of the *Ncor1* eleventh exon led to a 95.3% reduction (*p* < 0.05) in *Ncor1* transcript detection in the small intestine and 90.5% reduction (*p* < 0.05) in the colon (Figure 1A). *Apc^Min/+^*; *Ncor1^ΔIEC^* (mutants), *Apc^Min/+^*; *Ncor1^ΔIEC/+^* (heterozygotes), and *Apc^Min/+^; Ncor1* (control) littermate mice were sacrificed and the number of polyps was assessed in each individual along the rostrocaudal axis of the gut. An important effect on intestinal tumor initiation was observed in both heterozygous and mutant mice compared to that of the control mice (Figure 1B). Overall, the loss of one *Ncor1* allele in the intestinal epithelium led to a 29.3% reduction in polyp load in the *Apc^Min/+^* background (*p* < 0.05) and a 41.3% reduction when both *Ncor1* alleles were lost under the same background (*p* < 0.001) (Figure 1B). The polyp average diameter of all group of mice was not significantly different when compared together (Figure 1C). Overall, these observations supported that abrogation of intestinal epithelial NCOR1 expression reduced initiation of intestinal neoplasia under the *Apc^Min/+^* background without affecting polyp growth.

### 3.2. NCOR1 Is Required for CRC Cell Growth and Adherence

To further investigate the intrinsic epithelial role of NCOR1 in intestinal tumorigenesis, we depleted NCOR1 expression by RNA interference in both Caco-2/15 and HT-29 CRC cell lines. These cells were infected with lentiviruses expressing a non-target control shRNA (shNonTarget) or different shRNAs directed against NCOR1 transcripts (shNCOR1_654, 655, 656, and 657). Both cell lines were incubated with lentivirus particles for 48 h and then selected with puromycin for up to 7 d. Over the course of the first experiments conducted with Caco-2/15 and HT-29 cells, the morphology of NCOR1-depleted cells was very different from that of control cells. The cells were flattened, vacuolated, and their proliferation rate appeared to be reduced compared to that of the control cells. Thus, we monitored cell proliferation in these puromycin-selected populations by counting the number of cells on different days after an equivalent number of selected reseedings after selection. Caco-2/15 cells displayed a severe lack of proliferation when shNCOR1 constructs were stably introduced instead of control cells integrated with the shNonTarget construct (Figure 2A). The same phenotype was observed when HT-29 cells were transfected with the shNCOR1_655 construct (Figure 2B). qRT-PCR confirmed that this construct reduced the NCOR1 transcript by more than 80% in both Caco-2/15 (Figure 2C) and HT-29 (Figure 2D) cells. HT-29 shNCOR1_655 cells were unable to grow independent of anchorage instead of HT-29 shNonTarget cells (Figure 2E). Additionally, clonogenic assays showed that NCOR1-depleted CRC single cells did not retain the ability to form colonies compared to control cells (Figure 2F). These results suggest that NCOR1 is required for the growth of CRC cell lines under plastic and anchorage-independent conditions.

Next, we performed cell adhesion assays because NCOR1-depleted cells appeared to be less adherent during cell passages. After seeding 1 × 10^6^ cells per condition, we counted the number of cells in the culture medium after 16 h. We observed a significantly higher number of shNCOR1_655 cells in suspension than in control cells in Caco-2/15 and HT-29 cell lines (Figure 3A). To further assess the viability of adhered cells versus cells in suspension, we performed immunoblotting to detect the status of polyADP ribose polymerase-1 (PARP-1) protein. The appearance of PARP-1 fragments is widely accepted as a hallmark of programmed cell death [30]. Both the control and NCOR1-depleted adherent cell populations of each cell line were negative for the cleaved fragment of PARP-1. This suggests that the adhered NCOR1-depleted cells were still alive under these conditions. In contrast, nonadherent NCOR1-depleted Caco-2/15 and HT-29 cells displayed an apparition of the PARP-1 cleaved fragment, with levels comparable to the positive anoikis controls of cells seeded on polyHema-coated Petri dishes (Figure 3B).

### 3.3. Loss of NCOR1 Promotes CRC Cell Senescence Associated with a Secretory Phenotype

Since adherent NCOR1-depleted cells stopped proliferating without apparent signs of cell death, we hypothesized that they were senescent. We first performed immunoblotting to monitor NCOR1 protein expression compared to the histone variant γH2A.X phosphorylated on serine 139, a classic marker for senescent cells that becomes activated when DNA damage occurs [31]. NCOR1 protein levels were greatly reduced in Caco-2/-15 cells that expressed the shNCOR1_655 construct and led to an induction of the γH2A.X phosphorylated signal (Figure 4A). Next, we stained the cells with an X-Gal solution to visualize SA β-galactosidase activity. shNCOR1_655 directed against *NCOR1* led to the appearance of positive blue-stained cells in both Caco-2/15 and HT-29 cell lines (Figure 4B).

An additional feature of cellular senescence is the acquisition of a SASP. A conditioned culture medium from serum-deprived Caco-2/15, HT-29 shNCOR1_655, and shNonTarget cells was used to screen a human cytokine array panel to determine whether these proteins were modulated under these conditions. We determined that CXCL1, CXCL8, soluble intercellular adhesion molecule-1 (sICAM-1), and MIF were increased in shNCOR1_655 cells compared to the control cells (Figure 5A). ELISA confirmed that the loss of NCOR1 in Caco-2/15 cells caused a significant increase in CXCL1 (Figure 5B), CXCL8 (Figure 5C), MIF (Figure 5D), and sICAM-1 (Figure 5E) in the culture medium of these cells. Additionally, significant increases in CXCL1, CXCL8, and MIF were also observed in HT-29 shNCOR1_655 cells compared to their controls (Figure 5B–D). Overall, our observations indicated that upon the loss of NCOR1 expression, CRC cells exhibited a secretory phenotype associated with cellular senescence.

### 3.4. Loss of NCOR1 Reduces Tumorigenic Growth of HT-29 Cells In Vivo

Next, we performed xenograft experiments in nude mice with HT-29 cell lines that had integrated shNonTarget or shNCOR1_655. Following subcutaneous injection of viable cells into the dorsal region of the mice, we monitored tumor growth by measuring their length and width regularly (Figure 6). As expected, HT-29 control cells led to palpable tumors under these conditions (Figure 6A,B). Downregulation of NCOR1 in these cells drastically impaired the capacity of HT-29 cells to form tumors under these conditions (Figure 6A,B), with a coincident reduction in tumor weight at the time of mouse sacrifice compared to the control HT-29 cells (Figure 6C).

### 3.5. Impact of NCOR1 Depletion on CRC Cells Transcriptome

To gain insights into the transcriptional role of NCOR1 in CRC cells, we performed RNA-seq transcriptomic analyses in Caco-2/15 and HT-29 adherent cells depleted of NCOR1. After applying a cutoff of *p* < 0.05, we identified 2208 gene transcripts modulated at least twofold in Caco-2/15 shNCOR1_655 cell populations (57% upregulated; Figure 7A and Appendix A) and 2486 gene transcripts in HT-29 shNCOR1_655 cell populations (61% upregulated; Figure 7A and Appendix A) compared to their respective shNonTarget controls. A comparison of both analyses identified 421 common genes that were altered in both Caco-2/15 and HT-29 cell lines deficient in NCOR1 expression (Figure 7A, Appendix A). Of these genes, 62% were upregulated (Figure 7A, Appendix A). We then used the IPA software to generate signaling cellular pathway analyses of the common genes influenced by NCOR1 expression in each of these cell lines. EIF2 signaling, crucial to initiate translation and upregulated in CRC [32], was predicted to be downregulated in the absence of NCOR1 in both CRC cell lines (Figure 7B). In accordance with the anti-tumorigenic role of AMP-activated kinase (AMPK) [33], this pathway was also upregulated in the absence of NCOR1 (Figure 7B). Intriguingly, this analysis also predicted increases in several pathways positively involved in cellular migration and CRC metastasis signaling, including the RHO-GTPase pathway that contributes to cancer progression and metastasis [34] (Figure 7B). Several upstream activators of inflammatory processes, such as IFNγ, IL-1β, and TGF-β, as well as central inflammatory transcriptional regulators including NF-kβ and SMAD3, were also predicted to be activated in the absence of NCOR1 (Figure 7C). Additionally, our analysis supported the above findings by identifying several genes associated with cell senescence (Table 1) and SASP (Table 2). To further validate some of the targets associated with signaling cellular pathways as listed above, we investigated their gene transcript expression profiles following the knockdown of NCOR1 expression in both Caco-2/15 and HT-29 cells after 3 and 7 days, following introduction of shRNA constructs. The eukaryotic translation initiation factor 4B (*EIF4B*) and nucleophosmin 1 (*NPM1*) transcripts, which encode for RNA binding protein with roles in protein synthesis [35,36], were significantly reduced in both cell lines depleted for NCOR1 expression at all time points (Figure 8A,B). Immunity-related GTPase family M protein (*IRGM*), involved in autophagy, was induced with variable fold inductions depending on the time point (Figure 8C). Rap guanine nucleotide exchange factor 4 (*RAPGEF4*), a direct target of cAMP involved in activation of the GTPase Rap [37], was drastically induced in cells depleted for NCOR1 (Figure 8D). Brain-derived neurotrophic factor (*BDNF*), a growth factor with relevance in several cancer types [38], was also found strongly to be induced in the absence of NCOR1 (Figure 8E). CEA cell adhesion molecule 5 (*CEACAM5*) gene transcripts were also increased in absence of NCOR1, while this effect was not significant in HT-29 cells as opposed to Caco-2/15 cells (Figure 8F). Urothelial cancer associated 1 (*UCA1*), integrin subunit beta 6 (*ITGB6*), integrin subunit alpha 10 (*ITGA10*), rho GTPase activating protein 6 (*ARHGAP6*), and A-Kinase anchoring protein 12 (*AKAP12*), all involved in reorganization of the actin cytoskeleton [39,40,41], were all confirmed to be induced in the absence of NCOR1 (Figure 8G–K).

SOX2 is a reprogramming transcription factor associated with several cancer types, including CRC, and positively influences invasion/metastasis and resistance to cancer therapies [42]. We finally investigated the profile of SOX2 expression following the knockdown of NCOR1 expression in both Caco-2/15 and HT-29 cells. Significant induction of *SOX2* transcript expression was observed as early as 3 d following lentiviral infection in both cell lines with shNCOR1 constructs (Figure 9A). Immunoblots indicated that Caco-2/15 cells constitutively expressed SOX2 protein in contrast to HT-29 cells (Figure 9B). However, the knockdown of NCOR1 in both cell lines resulted in increased SOX2 protein levels under these conditions (Figure 9B).

## 4. Discussion

Intestinal neoplasia is a consequence of several molecular alterations that ultimately affect biological processes, such as cell proliferation. Due to the fact that senescence is initiated with an irreversible cessation of the cell cycle, it has long been considered a cellular means to protect against cancer development. However, senescent cells can harbor SASP features that stimulate a pro-inflammatory and tissue remodeling environment, leading to a higher risk of developing cancer. Our data suggest that NCOR1′s role in senescence could lead to anti- or pro-cancerous actions, depending on the context.

Although the literature has not yet established a clear association between NCOR1 and cellular senescence, some clues support this relationship. Drastic changes in chromatin structure can promote the establishment of senescence programs. In support of this, HDAC inhibitors have been reported to cause senescence in many cancer cell types, including CRC [43,44,45]. NCOR1 repressive activity is usually mediated via the recruitment of HDAC3 or other HDAC family members and, by extension, our observations are in line with these reports. Additionally, NCOR1 was identified as a central transcriptional repressor checkpoint of genes involved in inflammation and its deletion in macrophages resulted in the activation of several chemokines (CXCL-1, -2, -3, -4, -5, -8) and matrix metalloproteinase (MMP-13) encoding genes [3]. De-repression of each of these single genes, also associated with SASP, was observed in our NCOR1-depleted CRC cell lines (Table 2). A recent report showed that HDAC3 was required for non-small cell lung cancer cell tumorigenic growth, while central to the repression of p65 RelA/NF-κB-mediated induction of SASP in these cells [46]. Thus, our study is the first to functionally associate NCOR1 with cellular senescence, a process that most likely involves HDAC-associated repressive activities.

There are some indications that NCOR1 dysfunction may be associated with cancer. *NCOR1* is located on chromosome 17p11.2., in an area that is frequently mutated or lost in several human cancers. *NCOR1* is mutated in bladder cancer [47], breast cancer [48], and metastatic castration-resistant prostate cancer [49]. *NCOR1* point mutations have also been detected in two colorectal cancer cell lines [50]. In addition to mutations, aberrant cytoplasmic localization in NCOR1 has been described in CRC [51], retinoblastoma [52], and malignant melanoma [53]. Due to the fact that NCOR1 nuclear localization is essential to the repression of gene transcription, these observations support a loss of NCOR1 function, at least with regard to gene regulation. Although our results support that abrogation of NCOR1 function could be beneficial as a tumor suppressor strategy in CRC cells, our findings also point to the promotion of metastatic potential in the same context. These observations open up future patient-oriented investigations to delineate these paradoxical outcomes in this complex disease.

An important aspect of our work is the discovery of several CRC-associated genes, for which expression was influenced by the absence of NCOR1. NPM1 is involved in ribosome assembly and positively controls cell proliferation. NPM1 expression was reported to be increased in colon cancer and associated with lymph node metastasis [54]. Since NPM1 was found to be decreased in CRC cells depleted of NCOR1, this correlates well with the cell growth status of these cells. BDNF, a neurotrophin family member involved in brain development, has been reported to be increased in CRC [55] and to enhance CRC cell line motility [56]. Similarly, the long noncoding RNA UCA1 was found enriched in the serum exosomes of CRC patients with a positive outcome on CRC cell invasion and metastasis [57]. CEACAM5 adhesion molecules were reported to be overexpressed in many cancers, including CRC. The use of antibodies targeting these cell surface antigens was shown to be effective in reducing CRC cell invasion [58]. ITGB6 was detected in the serum of CRC patients, with a strong predictive index for the onset of metastatic state and tumor recurrence [59]. The Rho GTPase activating protein ARHGAP6 was also identified as being overexpressed in several CRC cell lines and tissues [60]. Due to the fact that Rho GTPases are crucial for the coordination of events required for cell migration [61], this correlates well with the predicted increase of metastatic potential in the absence of NCOR1. AKAP12 has also been described as being induced in colon cancer cells [62]. Another report showed that overexpression of AKAP12 in a CRC cell line inhibited proliferation and anchorage-independent growth, features that correlate well with our observations [63]. Based on these correlations, changes in expression of several identified NCOR1 gene targets suggest a functional role of this corepressor in promoting CRC cell growth while opposing metastasis potential under the same cellular context.

Our transcriptomic analysis also identified SOX2 as being upregulated in the absence of NCOR1. SOX2 is a transcriptional repressor that regulates several cellular processes associated with different types of cancer [42]. SOX2 expression has been associated with increased CRC metastasis [64], increased CRC cell migration [65], and cancer stem-like properties in CRC [66]. Some evidence also supports a role for SOX2 in reducing CRC cell proliferation via inhibition of the mTOR pathway [67]. Another report showed that the induction of SOX2 led to the onset of senescence in CRC cells [68]. These observations coincide well with our data and suggest a possible functional regulatory axis between NCOR1 and SOX2 in the control of cellular senescence. Interestingly, a recent study showed that the *SOX2* gene promoter was directly repressed from vitamin D receptor (VDR) occupancy on VDR elements in colon cancer cells [69]. VDR recruits NCOR1 to negatively regulate vitamin-D3-mediated transcription [70] and physical recruitment of NCOR1 on VDR gene targets was associated with altered repression of chromatin marks in prostate cancer [71]. Thus, it is tempting to speculate that *SOX2* could be directly targeted by NCOR1 via similar mechanisms. Finally, SOX2 is a pluripotency factor strongly associated with cancer stem cells (CSCs), with a poor prognosis in several cancers, including CRC [42,66]. Although cancerous senescent cells are non-proliferative by nature, it remains possible that the increase in SOX2 expression following NCOR1 depletion correlates with the development of dormant CSCs. There are some suggestions from the literature supporting the concept that dormancy in tumor cells that express stemness-associated genes can be acquired via senescence [72,73]. Future work will be required to dissect the biological and tumorigenic effects of this novel NCOR1-SOX2 regulatory axis during intestinal development and diseases.

## 5. Conclusions

Our study identified a novel role for NCOR1 in protecting CRC cells against the initiation of senescence and SASP. Although depletion of NCOR1 drastically inhibited tumorigenic growth in these cells, a pro-metastasis signature was also observed based on transcriptomic analyses. Our observations highlight the complexity of targeting senescence as an overall tumor suppression strategy, but open up possibilities for future studies to manipulate NCOR1 at different disease stages in a program designed to treat colon tumorigenesis.

## Figures and Tables

**Figure 1 cancers-13-04414-f001:**
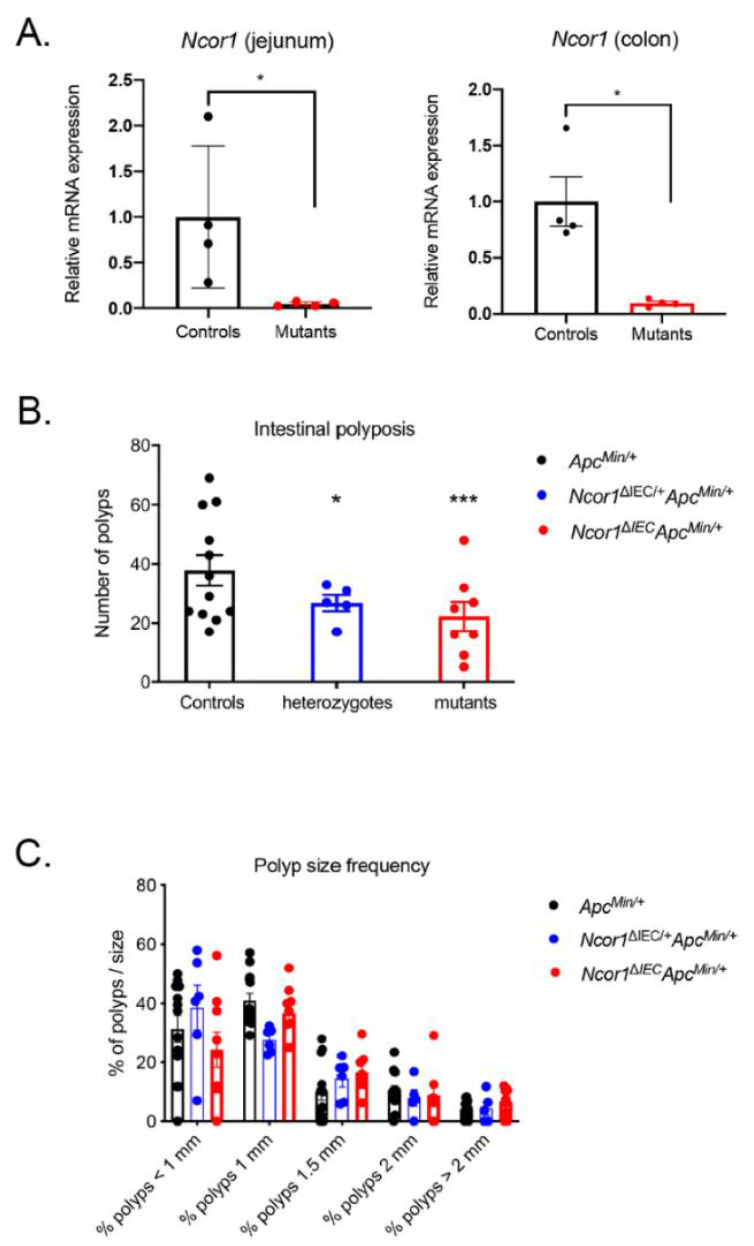
*Ncor1* epithelial deletion reduces polyposis in *Apc^Min/+^* mice. (**A**) Validation of the efficiency of *Ncor1* conditional intestinal deletion in mice by qRT-PCR analysis. *Ncor1* transcript expression was measured from the total RNA isolated from adult *Ncor1^ΔIEC^* and control mice (*n* = 4; means ± SEM). *Tbp* gene transcript levels were used as reference. Student’s *t*-tests were used to measure the significance of the results (* *p* < 0.05). Total polyp macrocount (**B**) and polyp size distribution (**C**) in the whole intestine of adult *Apc^Min/+^*; *Ncor1^ΔIEC^* (mutants, *n* = 9), *Apc^Min/+^; Ncor1^ΔIEC/+^* (heterozygotes, *n* = 6), and *Apc^Min/+^*; *Ncor1* (control, *n* = 13) littermate mice. Data are presented as means ± SEM. Two-way ANOVA tests coupled with the Grubbs test were performed to measure the significance of the results (* *p* < 0.05; *** *p* < 0.001).

**Figure 2 cancers-13-04414-f002:**
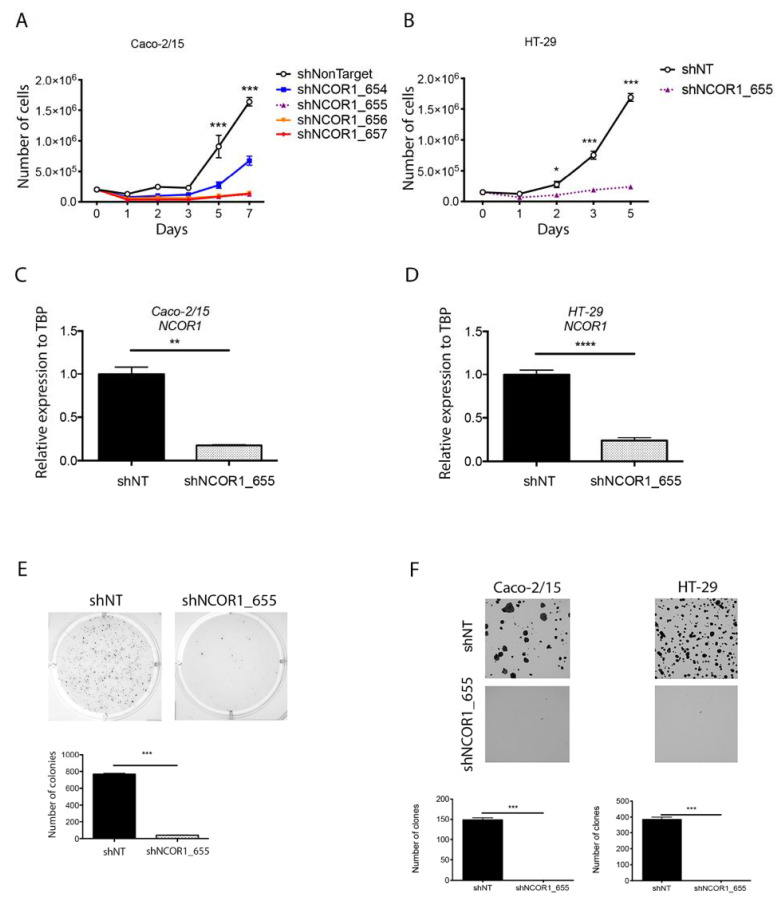
NCOR1 downregulation leads to slower proliferation with reduced tumorigenic potential of CRC cells ex vivo. (**A**) Validation of shRNAs directed against *NCOR1* by qRT-PCR analysis. *NCOR1* transcript expression was measured from total RNA isolated 7 days after infection from Caco-2/15 cells (*n* = 3; means ± SEM) and (**B**) HT-29 cells (*n* = 4; means ± SEM). TBP gene transcript levels were used as reference. Student’s *t*-tests were used to measure the significance of the results (* *p* < 0.05; *** *p* < 0.001). (**C**) Cell counts made 1, 2, 3, 5, and 7 days after seeding a fixed number of Caco-2/15 (** *p* < 0.01) and (**D**) HT-29 cells (*n* = 3; means ± SEM). Two-way ANOVA tests and a multiple comparison test by Bonferroni method were used to measure the significance of the results (**** *p* < 0.0001). (**E**) Photographs representative of an independent anchoring growth assay (soft agarose) in HT-29 cells. Image J software was used to count the number of colonies present in each of the conditions (*n* = 3; means ± SEM). Student’s *t*-test was used to measure significance (*** *p* < 0.001). (**F**) Clone formation assays performed in Caco-2/15 and HT-29 cells. Colonies were stained with crystal violet 14 days after seeding 1000 cells of each population. Image J software was used to count the number of clones present in each of the conditions (*n* = 3; means ± SEM). Student’s *t*-test was used to measure significance (*** *p* < 0.001).

**Figure 3 cancers-13-04414-f003:**
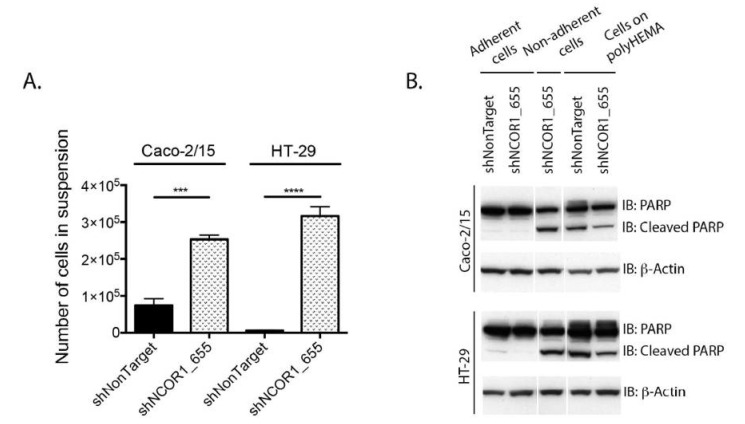
NCOR1 downregulation leads to reduced adhesion of CRC cells. (**A**) Total count of cells in suspension was done 16 h after seeding 1 × 10^6^ cells per condition (*n* = 3; means ± SEM). Student’s *t*-tests were used to measure the significance of the results (*** *p* < 0.001; **** *p* < 0.0001). (**B**) Immunoblots were performed to detect the complete and cleaved portions of PARP-1. Immunoblots were carried out with total protein extracts of Caco-2/15 and HT-29 cells obtained from plastic-adhered cells, cells in suspension (16 h after seeding), or cells that were seeded on petri dishes covered with polyHema (*n* = 3). The relative level of β-actin for each sample was assessed to control for protein loading.

**Figure 4 cancers-13-04414-f004:**
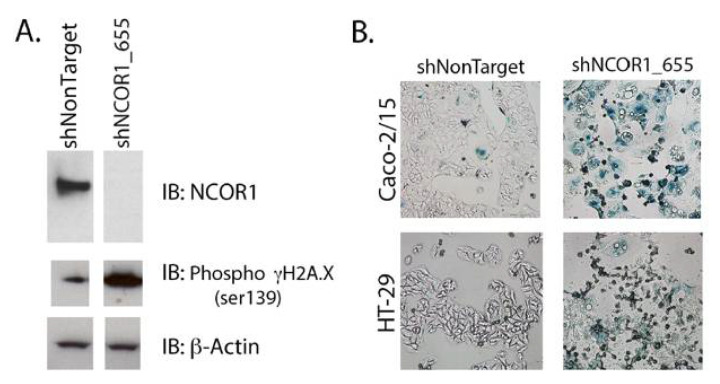
NCOR1 downregulation leads to cellular senescence in CRC cells. (**A**) Immunoblots were performed to detect NCOR1 protein and γH2A.X serine 139 phosphorylation. Total protein extracts from Caco-2/15 cells expressing a control shRNA (shNonTarget) or shNCOR1_655 (7 days after infection) were used (*n* = 3). The relative level of β-actin protein was monitored to control for protein loading. (**B**) SAβ-galactosidase staining was performed on Caco-2/15 (3.5 h) and HT-29 (8.5 h) cells expressing either shNonTarget or shNCOR1_655, 7 days after infection.

**Figure 5 cancers-13-04414-f005:**
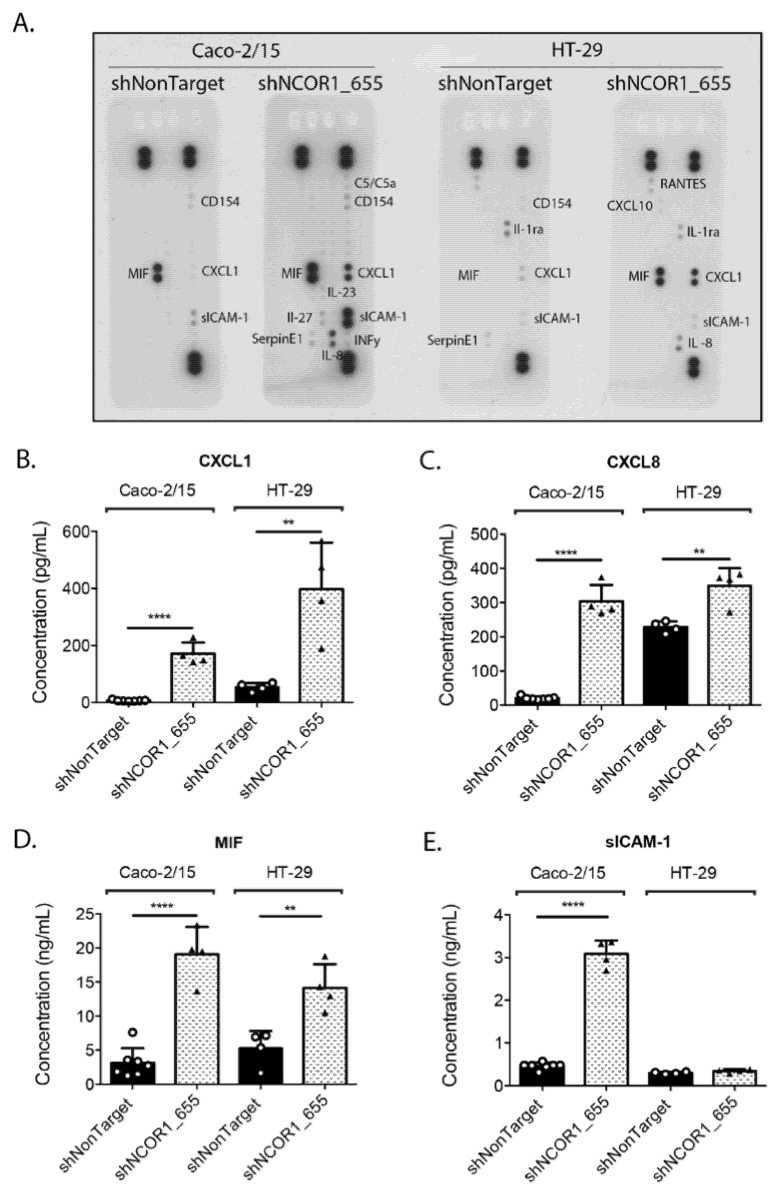
NCOR1 downregulation leads to a SASP in CRC cells. Caco-2/15 and HT-29 cells with integrated control shRNA (shNonTarget) or shNCOR1_655 were deprived of serum after 7 days of infection and the conditioned medium was then harvested after 48 h. Protein detection of cytokines and chemokines that had increased in SASP was visualized with the Human Cytokine Array Panel A kit (**A**). Specific ELISAs were used to quantify CXCL1 (**B**), CXCL8 (**C**), MIF (**D**), and sICAM-1 (**E**) (*n* = 3; means ± SEM). Student’s *t*-tests were used to measure the significance of the results (** *p* < 0.01; **** *p* < 0.0001).

**Figure 6 cancers-13-04414-f006:**
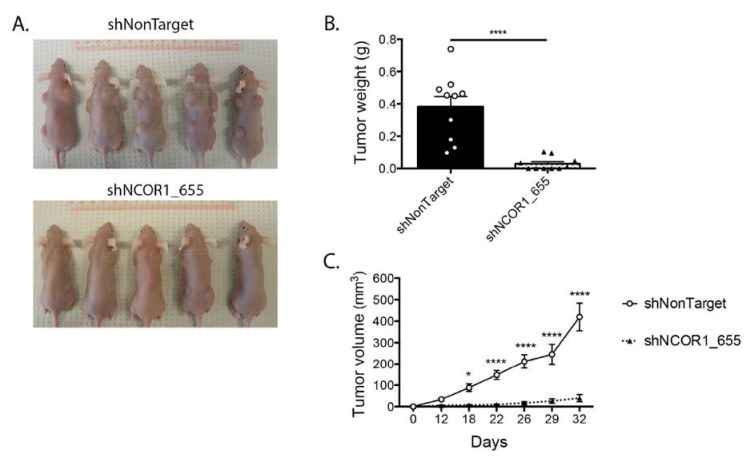
NCOR1 downregulation leads to a reduction of HT-29 tumorigenic cell growth in vivo. Xenografts were performed with female nu/nu mice aged between 4 to 6 weeks. (**A**) Representative photographs of mice 32 days after injection of HT-29 shNonTarget or shNCOR1_655 cells; 7 d after lentiviral infections, populations of viable cells were injected subcutaneously into the shoulder blades and hips of 5 animals per condition (1 × 10^6^ adhered cells per site resuspended in 0.1 mL DMEM, with four independent sites per mouse). (**B**) Tumor growth curves were established by measuring shoulder blade tumors lengthwise and widthwise using the formula (d2X D)/2 to calculate tumor volume. Two-way ANOVA tests and a multiple comparison test by Bonferroni’s method were used to measure the significance of the results (* *p* < 0.05; **** *p* < 0.0001). (**C**) Graph representing the mean weight of shoulder blade tumors at the time of sacrifice (*n* = 2 (2 × 5 mice per condition); means ± SEM). Student’s *t*-tests were used to measure the significance of the results (**** *p* < 0.0001).

**Figure 7 cancers-13-04414-f007:**
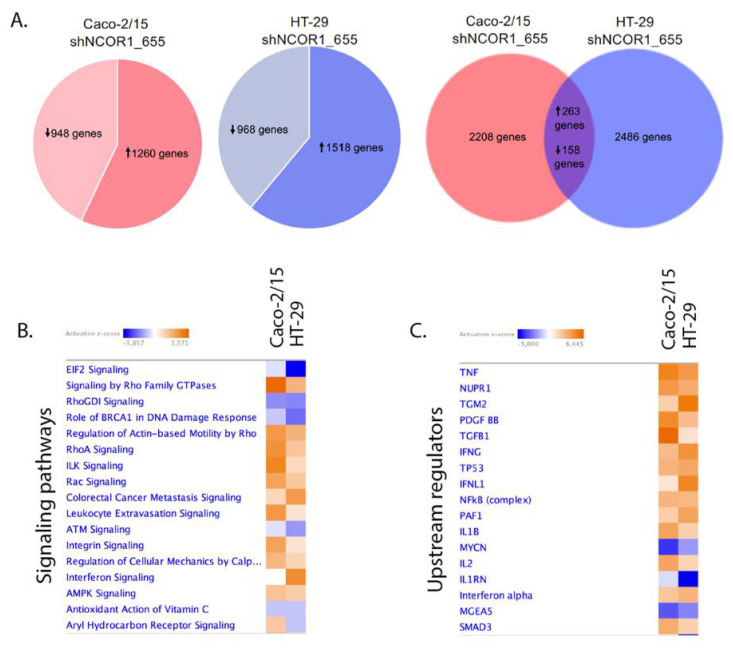
Comparative analysis of genes significantly modulated in Caco-2/15 and HT-29 cells depleted for NCOR1 expression. Total RNA from shNCOR1_655 and shNonTarget adherent cells was isolated 7 days after lentiviral infection and RNA-seq was performed. (**A**) A Venn diagram is used to display comparisons of gene transcript levels that were significantly modulated more than twofold (*p* < 0.05) between Caco-2/15 and HT-29 cells. IPA analysis showing altered signaling pathways (**B**) or upstream regulators (**C**) based on differentially expressed genes associated with the loss of NCOR1 expression in both Caco-2/15 and HT-29 cells. The color range indicates predicted activation state where a value of z-score ≥ 2 corresponds to a significant prediction of activation (orange), whereas a value ≤ −2 is associated with inhibition (blue).

**Figure 8 cancers-13-04414-f008:**
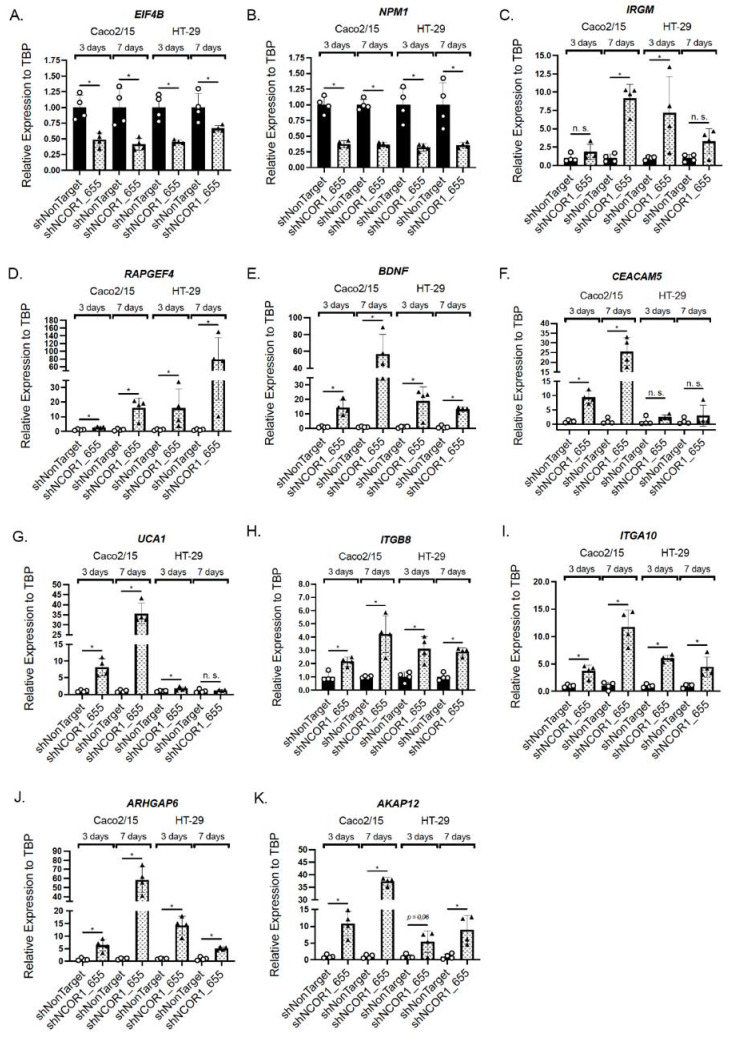
Confirmatory analysis of NCOR1 gene targets in CRC cells. Total RNA was isolated at 3 and 7 days following lentiviral infection of Caco-2/15 and HT-29 cells with shNonTarget or shNCOR1_655 constructs. *EIF4B* (**A**), *NPM1* (**B**), *IRGM* (**C**), *RAPGEF4* (**D**), *BDNF* (**E**), *CEACAM5* (**F**), *UCA1* (**G**), *ITGB8* (**H**), *ITGA10* (**I**), *ARHGAP6* (**J**), and *AKAP12* (**K**) expression values relative to *TBP* were measured by qRT-PCR (*n* = 3–4; means ± SEM). Student’s *t*-tests were used to measure the significance of the results (* *p* < 0.05).

**Figure 9 cancers-13-04414-f009:**
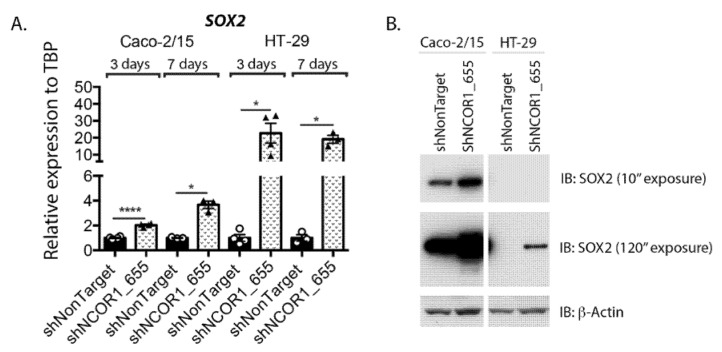
NCOR1 downregulation increases SOX2 expression in CRC cells. (**A**) Total RNA was isolated at 3 and 7 days following lentiviral infection of Caco-2/15 and HT-29 cells with shNonTarget or shNCOR1_655 constructs. *SOX2* expression relative to *TBP* was measured by qRT-PCR (*n* = 3–4; means ± SEM). Student’s *t*-tests were used to measure the significance of the results (* *p* < 0.05; **** *p* < 0.0001). (**B**) Immunoblots were performed to detect SOX2 protein expression. Total protein extract was isolated at 7 days following lentiviral infection of Caco-2/15 and HT-29 cells with shNonTarget, shNCOR1_655, or shNCOR1_656 constructs. The relative level of β-actin protein was monitored to control protein loading (*n* = 3).

**Table 1 cancers-13-04414-t001:** List of genes associated with cell senescence and significantly modulated (change ratio of >2; *p* < 0.05) in both Caco-2/15 and HT-29 cells depleted for NCOR1 expression.

Symbol	Gene Name	Caco-2/15		HT-29	
		shNCOR1_655		shNCOR1_655	
		Ratio	*p*-Value	Ratio	*p*-Value
BHLHE40	basic helix-loop-helix family, member e40	2.955	1.10 × 10^−46^	4.211	6.70 × 10^−10^
BMP7	bone morphogenetic protein 7	−2.123	1.20 × 10^−24^	−1.134	1.00 × 10^+00^
BTG3	BTG family, member 3	2.097	4.20 × 10^−23^	1.119	5.20 × 10^−01^
CDK6	cyclin-dependent kinase 6	2.563	2.90 × 10^−40^	4.179	9.10 × 10^−23^
CDKN1A	cyclin-dependent kinase inhibitor 1A (p21, Cip1)	3.394	2.90 × 10^−41^	1.485	5.60 × 10^−01^
CDKN1C	cyclin-dependent kinase inhibitor 1C (p57, Kip2)	−9.363	2.20 × 10^−11^	−1.297	6.00 × 10^−01^
CDKN2B	cyclin-dependent kinase inhibitor 2B (p15, inhibits CDK4)	2.032	6.00 × 10^−04^	3.138	1.80 × 10^−02^
CDKN2D	cyclin-dependent kinase inhibitor 2D (p19, inhibits CDK4)	2.521	7.80 × 10^−04^	1.397	2.90 × 10^−02^
EREG	epiregulin	3.961	3.00 × 10^−51^	2.328	9.70 × 10^−02^
IFI16	interferon, gamma-inducible protein 16	3.758	5.50 × 10^−04^	2.495	1.10 × 10^−02^
JUNB	jun B proto-oncogene	2.21	1.20 × 10^−20^	1.258	5.10 × 10^−01^
NPM1	nucleophosmin (nucleolar phosphoprotein B23, numatrin)	−2.417	2.10 × 10^−36^	−3.308	1.90 × 10^−18^
RB1	retinoblastoma 1	−3.057	1.20 × 10^−50^	−4.202	1.90 × 10^−25^
SKP1	S-phase kinase-associated protein 1	−3.078	6.40 × 10^−56^	−3.156	6.30 × 10^−17^
SOX2	SRY (sex determining region Y)-box 2	2.39	5.80 × 10^−06^	20.563	1.30 × 10^−10^
TERT	telomerase reverse transcriptase	−3.143	2.50 × 10^−04^	−2.942	2.90 × 10^−02^
TGFBR1	transforming growth factor, beta receptor 1	−2.844	1.10 × 10^−44^	−4.356	2.70 × 10^−23^
TM4SF1	transmembrane 4 L six family member 1	2.646	2.30 × 10^−06^	2.152	8.00 × 10^−09^
TP73	tumor protein p73	−4.031	2.30 × 10^−24^	−4.646	5.20 × 10^−08^
VCAN	versican	9.747	3.70 × 10^−15^	2.944	3.60 × 10^−02^

**Table 2 cancers-13-04414-t002:** List of genes associated with SASP and significantly modulated (change ratio of >2; *p* < 0.05) in both Caco-2/15 and HT-29 cells depleted for NCOR1 expression.

Symbol	Gene Name	Caco-2/15		HT-29	
		shNCOR1_655		shNCOR1_655	
		Ratio	*p*-Value	Ratio	*p*-Value
CSF1	colony stimulating factor 1 (macrophage)	3.42	6.70 × 10^−11^	7.301	2.40 × 10^−43^
CXCL1 (GRO-a)	chemokine (C-X-C motif) ligand 1 (melanoma growth stimulating activity, alpha)	2.576	1.00 × 10^−04^	8.168	3.90 × 10^−14^
CXCL2 (GRO-b)	chemokine (C-X-C motif) ligand 2	2.385	8.10 × 10^−08^	2.774	2.70 × 10^−02^
CXCL3	chemokine (C-X-C motif) ligand 3	4.124	3.10 × 10^−05^	3.88	7.20 × 10^−05^
CXCL5	chemokine (C-X-C motif) ligand 5	26.337	7.40 × 10^−04^	15.095	6.50 × 10^−02^
EREG	epiregulin	3.961	3.00 × 10^−51^	2.328	9.70 × 10^−02^
FASLG	Fas ligand (TNF superfamily, member 6)	36.277	2.80 × 10^−02^	1.012	1.00 × 10^+00^
FN1	fibronectin 1	2.418	7.80 × 10^−13^	1.421	8.20 × 10^−01^
ICAM1	intercellular adhesion molecule 1	2.299	2.10 × 10^−16^	3.304	2.60 × 10^−10^
ICAM3	intercellular adhesion molecule 3	−2.218	6.20 × 10^−14^	−3.247	1.20 × 10^−08^
IGFL2	IGF-like family member 2	4.942	4.00 × 10^−19^	4.554	2.00 × 10^−07^
IL15	interleukin 15	5.476	7.80 × 10^−10^	2.072	1.40 × 10^−06^
ITPKA	inositol-trisphosphate 3-kinase A	−3.24	1.90 × 10^−20^	−4.127	3.30 × 10^−01^
MMP13	matrix metallopeptidase 13 (collagenase 3)	14.065	9.60 × 10^−04^	5.692	7.80 × 10^−05^
MMP14	matrix metallopeptidase 14 (membrane-inserted)	2.858	1.00 × 10^−26^	1.999	2.80 × 10^−02^
TIMP1	TIMP metallopeptidase inhibitor 1	2.186	3.90 × 10^−02^	1.861	5.00 × 10^−06^

## Data Availability

RNA-seq data were deposited in the Gene Expression Omnibus database (GSE174204).

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
