# Peer review of "NCOR1 Sustains Colorectal Cancer Cell Growth and Protects against Cellular Senescence"

_cancers, 2021, doi:10.3390/cancers13174414_

Round 1
Reviewer 1 Report
The authors provided significant changes to improve this paper.
Reviewer 2 Report
I understand that most of the points i raised in the revision process would need far more time to obtain as the breeding for additional mouse individuals would take several months.
Therefore, I would recommend accepting the manuscript in its current form.
Reviewer 3 Report
The manuscript shows the potential role of NCOR1 in the CRC progression. Authors use shNCOR1 strategies of silencing to support their working hypothesis.
Data are clearly presented. The main weakness is the lack of validation in CRC human samples and xenografts. However, as an initial step, the use of the cell lines are appropriate for the analysis.
Minor: revise chemical formulations (sub-index).
Reviewer 4 Report
I consider that you have addressed all the comments raised by Reviewer 1
This manuscript is a resubmission of an earlier submission. The following is a list of the peer review reports and author responses from that submission.
Round 1
Reviewer 1 Report
In this work St-Jean et al. investigate the role of the repressor NCOR1 in preventing senescence of colon cancer cells. They also look into the dual role of this process in both mitigating and inducing tumorigenesis.
The research is well-conceived and the paper well written but some gaps need to be filled. In particular, the last part of the work gives the impression of a missed opportunity to deepen the RNA-seq data presented.
The following comments are meant to improve the paper and make it more comprehensible.
MAJOR POINTS
- lines 257-263: authors describe the great sensitivity of both cell lines to constitutive NCOR1-KD (yet reported in Materials and Methods, lines 122-124). Given these morphological changes and fast cell line loss due to massive growth inhibition, maybe an inducible system (such as pLKO system) would be more appropriated and enable authors to better and easier study the effects of NCOR1 KD.
- Figure 2F is just qualitative, a graph reporting data from more than one experiments is needed
- lines 303-308: to investigate why NCOR1-KD cells slightly adhere to the substrate, authors analyze PARP-1 status to evaluate cell death. This evaluation could also be done by cell cycle analysis, which will also assess if cells directly undergo apoptosis or are blocked in a different cell phase before. Moreover, since adhesion strongly relies on specific molecules on the cell surface. It would be interesting to investigate a panel of these molecules to evaluate if they partake in this phenotype.
- there no consistency in the use of the different shRNA targeting NCOR1. In figure 2, experiments suggest that shRNA-655 is the best performing and, after an initial screening, authors chose to go on whit that one. This perfectly makes sense except for other shRNAs re-appear later. In Figure 3, panel A report only shRNA-655 while panel B show also shRNA-656. In Figure 4 there are again all the shRNAs. In figure 8, panel A show only shRNA-655 while panel B have also shRNA-656 which, besides, do not properly work, so why authors show it? All these variances are quite confusing. If all experiment (at least the once up to Figure 4) cannot be performed with all shRNAs, thus authors should remove results obtained with all but shRNA-655 after the first evaluation in Figure 2.
- line 317-318: “…significant change in their morphology…”, stating it for the third time now authors may want to show images of this altered morphology
- lines 388-389: “In accordance with the anti-tumorigenic role of AMP-activated kinase (AMPK)…” not questioning the relevant role of AMPK in carcinogenesis, but Figure 7B show many more interesting pathways strongly up-regulated (even more than AMPK) in both cell lines. In particular, RHO-GTPase pathway is the second top-scoring and it is not even mentioned. Authors are invited to discuss RNA-seq results more carefully.
- line 405: “…Caco-2/15 cells constitutively expressed SOX2…” this observation implies that Caco-2/15 are NOT a good model to study the relationship between NCOR1-KD and subsequentially SOX2 re-expression. Generally, the choice of focusing on SOX2 is not clear and not strongly supported by data. It is not the top-scoring up-regulated gene in both cell lines (7th in HT29 and 198th in Caco) and, as said before, its relationship with NCOR1 is not confirmed in Caco. Moreover, the authors do not provide any phenotypic experiment (at least proliferation, colony formation, adhesion, senescence assays) to evaluate if SOX2 over-expression could mimic, at least in part, the effect of NCOR1 KD. Actually, this last chapter does not enrich the whole paper since it is just an incomplete analysis of the role of SOX2. If further experiments on SOX2 cannot be performed, maybe authors could analyze the effect of other NCOR1 target genes or, at least, discuss more in-depth the gene target resulted from the RNA-seq.
- By the end of the Discussion and in the Conclusion chapter, it feels like something is missing. As said before, readers can be confused by the focus on SOX2 in the discussion/conclusion without considering ANY of the other NCOR1 target genes and without proof of SOX2 strong connection with the biological effects described so far. If authors do not wish to analyze/discuss more in-depth other targets, they could at least focus on another interesting part of their results: SASP role in counteracting NCOR1 anti-tumor effects could be fascinating and new but it is only cited in the Conclusion.
MINOR POINTS
- lines 67-68: “… ranking third and second in terms of mortality…”, third OR second?
- lines 87-88: authors state that they will perform experiments in ApcMin/+ mice, but they do not explain why. If this detail is so important to be specified by the end of the Introduction chapter, thus maybe it needs a little clarification. Moreover, in lines 52-65, authors describe studies on the role of NCOR1 performed in mice but never cite the strain they will use thus failing the connection between these two part of the introduction
- Figure 2 (particularly panels A-B) will be clearer with coloured lines
- line 318: “..displayed no apparent sign of cell death.”, just a few lines above (302-303) authors say that “The appearance of PARP-1 fragments is widely accepted as a hallmark of programmed cell death” and then demonstrate, with figure 3B, that shNCOR1 cells have cleaved PARP.
- line 324: “…signal was coincidently observed…” what does COINCIDENTLY means? It sounds not very scientific and it is not further explained.
- Figure 7C is not properly described in the main text. The notion of “upstream regulators” is not even mentioned, thus making this panel unclear and irrelevant
- lines 436-437: “Our data support the concept of dual roles for senescence in anti- or pro-cancerous actions.” No, data presented here are about the role of NCOR1
- lines 442-444: “NCOR1 repressive activity can be mediated via the recruitment of HDAC3 or other HDAC family members, and our observations are in accordance with these reports.” No, data presented here deals with HDACs or chromatin structure
- lines 464-466: “Our attempt to assess whether these cells can invade (in vitro assays) or metastasize (in vivo xenografts) in the absence of NCOR1 was not conclusive.” This is VERY confusing. In the Results chapter, data in Figures 2E-F, 3A and 6 are presented and described as significant but now “not conclusive”? Please, re-write this sentence making it consistent with what presented so far
- lines 476-477: “Interestingly, a recent study showed that the SOX2 gene promoter was directly repressed from vitamin D receptor (VDR) occupancy on VDR elements in colon cancer cells.” VDR is out of context. It is cited only in this sentence and is not properly addressed
Author Response
We would like to thank the reviewers for their comments as well as their overall shared enthusiasm on the submitted work. Please find a point-by-point discussion of the issues raised by the reviewers.
Reviewer 1:
MAJOR POINTS
- lines 257-263: authors describe the great sensitivity of both cell lines to constitutive NCOR1-KD (yet reported in Materials and Methods, lines 122-124). Given these morphological changes and fast cell line loss due to massive growth inhibition, maybe an inducible system (such as pLKO system) would be more appropriated and enable authors to better and easier study the effects of NCOR1 KD.
This is a good suggestion that will be useful to consider in the future. Nevertheless, constitutive knockdown of NCOR1 allowed us to obtain reproducible observations on NCOR1 functions in the context of 2 CRC cell lines.
- Figure 2F is just qualitative, a graph reporting data from more than one experiments is needed
Every single experiment that was performed in the context of figure 2f resulted each time in no colonies to be counted for each NCOR1-KD HT-29 and Caco-2/15 cells. Because of the null value, this prevented us to calculate a fold-reduction as a calibration between each independent experiment (n=3 with different cell populations each time as indicated in the figure legend). We felt that a graph would not be representative of our statement that “NCOR1-depleted CRC single cells did not retain the ability to form colonies compared to control cells” in the result section.
- lines 303-308: toinvestigate why NCOR1-KD cells slightly adhere to the substrate, authors analyze PARP-1 status to evaluate cell death. This evaluation could also be done by cell cycle analysis, which will also assess if cells directly undergo apoptosis or are blocked in a different cell phase before. Moreover, since adhesion strongly relies on specific molecules on the cell surface. It would be interesting to investigate a panel of these molecules to evaluate if they partake in this phenotype.
These are interesting alternative experiments to be considered in future work.
- there no consistency in the use of the different shRNA targeting NCOR1. In figure 2, experiments suggest that shRNA-655 is the best performing and, after an initial screening, authors chose to go on whit that one. This perfectly makes sense except for other shRNAs re-appear later. In Figure 3, panel A report only shRNA-655 while panel B show also shRNA-656. In Figure 4 there are again all the shRNAs. In figure 8, panel A show only shRNA-655 while panel B have also shRNA-656 which, besides, do not properly work, so why authors show it? All these variances are quite confusing. If all experiment (at least the once up to Figure 4) cannot be performed with all shRNAs, thus authors should remove results obtainedwith all but shRNA-655 after the first evaluation in Figure 2.
As requested, figures 3, 4 and 8 has been cropped to represent shRNA-656 data only.
- line 317-318: “…significant change in their morphology…”, stating it for the third time now authors may want to show images of this altered morphology
Unfortunately, we did not collect publishable images at the time the experiments were performed. We have modified our sentence at lines 308-309 to not overstate this observation: “Since adherent NCOR1-depleted cells stopped proliferating without apparent sign of cell death, we hypothesized that they were senescent.”
- lines 388-389: “In accordance with the anti-tumorigenic role of AMP-activated kinase (AMPK)…” not questioning the relevant role of AMPK in carcinogenesis, but Figure 7B show many more interesting pathways strongly up-regulated (even more than AMPK) in both cell lines. In particular, RHO-GTPase pathway is the second top-scoring and itis not even mentioned. Authors are invited to discuss RNA-seq results more carefully.
We have included this sentence (lines 402-405): “….in several pathways positively involved in cellular migration and CRC metastasis signaling, including the RHO-GTPase pathway that contribute to cancer progression and metastasis 34 »
- line 405: “…Caco-2/15 cells constitutively expressed SOX2…” this observation implies that Caco-2/15 are NOT a good model to study the relationship between NCOR1-KD and subsequentially SOX2 re-expression. Generally, the choice of focusing on SOX2 is not clear and not strongly supported by data. It is not the top-scoring up-regulated gene in both cell lines (7th in HT29 and 198th in Caco) and, as said before, its relationship with NCOR1is not confirmed in Caco. Moreover, the authors do not provide any phenotypic experiment (at least proliferation, colony formation, adhesion, senescence assays) to evaluate if SOX2 over-expression could mimic, at least in part, the effect of NCOR1 KD. Actually, this last chapter does not enrich the whole paper since it is just an incomplete analysis of the role of SOX2. If further experiments on SOX2 cannot be performed, maybe authors could analyze the effect of other NCOR1 target genes or, at least, discuss more in-depth the gene target resulted from the RNA-seq.
We respectfully disagree with this comment. Even if Caco-2/15 cells express SOX2, it does not mean that this is not a good model to show that absence of NCOR1 does impact on SOX2 expression, which we are in fact demonstrating. A published study, cited in our discussion, has used the parental Caco2 cell line and HT29 cells to demonstrate the involvement of SOX2 in the regulation of cell proliferation (Takeda et al., Sci Rep, 2018). In addition, SOX2 is commonly up-regulated in both CRC cell lines made deficient in NCOR1, which was an important criteria to focus on this gene among others. Additionally, as discussed in our manuscript, SOX2 is an important regulator of cancer stem cells and some evidence point for a role of this regulator in senescence and metastasis potential (Zheng et al., Oncotarget, 2017; Neumann et al., BMC Cancer, 2011). We have made a choice of investigating this target because several groups have performed experiments with SOX2 in CRC cell lines with some overlapping biological effects. Analyzing other targets is limitless given the number of commonly influenced targets in these two cell lines and the rationale of prioritizing any other targets would be similar to what we have decided to do for SOX2.
- By the end of the Discussion and in the Conclusionchapter, it feels like something is missing. As said before, readers can be confused by the focus on SOX2 in the discussion/conclusion without considering ANY of the other NCOR1 target genes and without proof of SOX2 strong connection with the biological effects described so far. If authors do not wish to analyze/discuss more in-depth other targets, they could at least focus on another interesting part of their results: SASP role in counteracting NCOR1 anti-tumor effects could be fascinating and new but it is only cited in the Conclusion.
As stated in the last comment, SOX2 has been already reported to mimic biological effects in CRC cell lines as we have observed (cell proliferation, senescence, metastasis potential, etc.). In our opinion, discussion of additional targets (we have identified more than 400 NCOR1 common targets in both CRC cell lines) could be endless and unfocused in this current manuscript.
MINOR POINTS
- lines 67-68: “… ranking third and second in terms of mortality…”, third OR second
This has been corrected as followed: “ranking third in incidence and second in terms of mortality…”
- lines 87-88: authors state that they will perform experiments in ApcMin/+mice, but they do not explain why. If this detail is so important to be specified by the end of the Introduction chapter, thus maybe it needs a little clarification. Moreover, in lines 52-65, authors describe studies on the role of NCOR1 performed in mice but never cite the strain they will use thus failing the connection between these two part of the introduction
The sentence referring the the ApcMin mice has been removed in the introduction section and the strain of NCOR1 deleted mice is now cited in the beginning of the result section (lines 229-230): “….generated mice lacking intestinal epithelial NCOR1 as previously described 8, but here into a C57BL/6J ApcMin/+ background”
- Figure 2 (particularly panels A-B) will be clearer with coloured lines
Coloured lines has been included in figure 2A and B as requested.
- line 318: “..displayed no apparent sign of cell death.”, just a few lines above (302-303) authors say that “The appearance of PARP-1 fragments is widely accepted as a hallmark of programmed cell death” and then demonstrate, with figure 3B, that shNCOR1 cells have cleaved PARP.
This was meant to refer to adherent cells, this is now better specified (lines 308-309) as followed: “Since adherent NCOR1-depleted cells stopped proliferating without apparent sign of cell death…”
- Line 324: “…signal was coincidently observed…” what does COINCIDENTLY means? It sounds not very scientific and it is not further explained.li
This sentence has been corrected as followed (lines 312-314): “NCOR1 protein levels were greatly reduced in Caco-2/-15 cells that expressed the shNCOR1_655 construct and led to an induction of the gH2A.X phosphorylated signal.”
- Figure 7C is not properly described in the main text. The notion of “upstream regulators” is not even mentioned, thus making this panel unclear and irrelevant
We are now providing with a short description of the data as followed (lines 405-407): “Several upstream activators of inflammatory processes such as IFNg, IL-1b and TGF-b as well as central inflammatory transcriptional regulators including NF-kb and SMAD3 were also predicted to be activated in the absence of NCOR1”
- lines 436-437: “Our data support the concept of dual roles for senescence in anti- or pro-cancerous actions.” No, data presented here are about the role of NCOR1
We have corrected this sentence as followed (lines 481-482): “Our data suggest that NCOR1’s role in senescence could lead to anti- or pro-cancerous actions depending on the context.”
- lines 442-444: “NCOR1 repressive activity can be mediated via the recruitment of HDAC3 or other HDAC family members, and our observations are in accordance with these reports.” No, data presented here dealswith HDACs or chromatin structure
We have modified our sentence as followed (lines 487-489): “NCOR1 repressive activity is usually mediated via the recruitment of HDAC3 or other HDAC family members and by extension, our observations are in line with these reports”
- lines 464-466: “Our attempt to assess whether these cells can invade (in vitro assays) or metastasize (in vivo xenografts) in the absence of NCOR1 was not conclusive.” This is VERY confusing. Inthe Results chapter, data in Figures 2E-F, 3A and 6 are presented and described as significant but now “not conclusive”? Please, re-write this sentence making it consistent with what presented so far.
We were trying to explain that we were not able to assess the metastasis potential of these cells by assays we have tried to conduct (data not shown). For the sake of not being confusing, we have removed this sentence.
- lines 476-477: “Interestingly, a recent study showed that the SOX2 gene promoter was directly repressed from vitamin D receptor (VDR) occupancy on VDR elements in colon cancer cells.” VDR is out of context. It is cited only in this sentence and is not properly addressed
NCOR1 is a corepressor that represses transcription via its recruitment from nuclear receptors. VDR is a nuclear receptor previously showed to recruit NCOR1 to repress gene transcription. We have adjusted this sentence and added one additional reference in order to clarify this fact for non-expert in the field (lines 520-525): “…..occupancy on VDR elements in colon cancer cells [52]. VDR recruits NCOR1 to negatively regulate vitamin D3-mediated transcription [53] and physical recruitment of NCOR1…..”
Reviewer 2 Report
The manuscript of St-Jean et al. thoroughly analyses the role of NCOR1 in colorectal cancer in-vitro and in-vivo. Conditional intestinal epithelial deletion of Ncor1 in ApcMin/+ mice resulted in a significant reduction in polyposis. NCOR1 downregulation in-vitro induced reduced cell growth associated with features of senescence, which was supported by RNA-seq data. Tumor growth of HT-29 cells was reduced in the absence of NCOR1 in mouse xenografts. According to RNA-seq analyses and in-vitro data SOX2-Expression was induced after NCOR1 knockdown.
The present study is carried out well and the conclusions are adequately drawn. I would recommend accepting the manuscript after the following revisions:
- The authors could provide immunohistochemical figures analyzing Ncor1-expression pattern in (murine) wild-type mucosa of jejunum and colon under special consideration where Ncor1 expression is located (epithelium?, stroma?, both?). In addition, it might be informative to illustrate the efficiency of the Ncor1-knock out in immunohistochemistries of epithelial jejunum and colon tissue (simultaneously stained and compared to wild-type tissue).
- Did the polyps in Ncor1-knock out mice show enhanced (epithelial?) SOX2-expression in jejunum and colon compared to wild-type tumors? Did the authors observe polyps with invasive tumor growth in histology or even distant metastasis (liver?)? Did the wild-type polyps show differential Ncor1 expression compared to surrounding non-neoplastic epithelium?
- It would be very interesting to see a histological image of the xenografts. Did the Ncor1-down-regulated tumors show a different (dys-cohesive?) growth compared to shNonTarget? Were tumor cell nuclei of shNCOR1_655 alive or necrotic?
- Lines 376-377: „To gain insights into the transcriptional role of NCOR1 in CRC cells, we performed RNA-seq transcriptomic analyses in both Caco-2/15 and HT-29 adherent cells that were reduced or not for NCOR1 expression“ Especially the last part of this sentence („that were reduced or not for NCOR1 expression“) might profit from a clearer formulation.
Author Response
We would like to thank the reviewers for their comments as well as their overall shared enthusiasm on the submitted work. Please find a point-by-point discussion of the issues raised by the reviewers.
- The authors could provide immunohistochemical figures analyzing Ncor1-expression pattern in (murine) wild-type mucosa of jejunum and colon under special consideration where Ncor1 expression is located (epithelium?, stroma?, both?). In addition, it might be informative to illustrate the efficiency of the Ncor1-knock out in immunohistochemistries of epithelial jejunum and colon tissue (simultaneously stained and compared to wild-type tissue).
We agree that such IH (or IF) would have been interested to perform. Unfortunately, our numerous attempts to perform such experiments with the use of different NCOR1 commercial antibodies were unsuccessful to obtain specific signals in the mouse intestine.
- Did the polyps in Ncor1-knock out mice show enhanced (epithelial?) SOX2-expression in jejunum and colon compared to wild-type tumors? Did the authors observe polyps with invasive tumor growth in histology or even distant metastasis (liver?)? Did the wild-type polyps show differential Ncor1 expression compared to surrounding non-neoplastic epithelium?
These are interesting questions that would be interesting to investigate in the future. Unfortunately, we did not surgically remove polyps from the ApcMin/Ncor1 compound mice to this end and as stated above, NCOR1 staining was not successful by IH or IF. Since additional mouse individuals with the Apc/Ncor1 deleted background would require several months to obtain, we would not be able to experimental address these interesting suggestions.
- It would be very interesting to see a histological image of the xenografts. Did the Ncor1-down-regulated tumors show a different (dys-cohesive?) growth compared to shNonTarget? Were tumor cell nuclei of shNCOR1_655 alive or necrotic?
Since the NCOR1 depleted HT-29 cells did not produce visible tumors in most xenograft experiments, we did not attempt to isolate biological material in these experiments. Performing such an analysis would require to repeat these experiments with the inclusion of many more samples in order to satisfactorily address if such biological differences occur significantly.
- Lines 376-377: „To gain insights into the transcriptional role of NCOR1 in CRC cells, we performed RNA-seq transcriptomic analyses in both Caco-2/15 and HT-29 adherent cells that were reduced or not for NCOR1 expression“ Especially the last part of this sentence („that were reduced or not for NCOR1 expression“) might profit from a clearer formulation.
We have clarified the sentence as followed (lines 388-390): To gain insights into the transcriptional role of NCOR1 in CRC cells, we performed RNA-seq transcriptomic analyses in Caco-2/15 and HT-29 adherent cells depleted of NCOR1.
Round 2
Reviewer 1 Report
I really appreciate the answers of the authors and their effort to improve and clarify this work. Anyway, some issue remains to be addressed. For some points, the explanation provided are not exhaustive and do not solve problems of clarity and consistency.
- Every single experiment that was performed in the context of figure 2f resulted each time in no colonies to be counted for each NCOR1-KD HT-29 and Caco-2/15 cells. Because of the null value, this prevented us to calculate a fold-reduction as a calibration between each independent experiment (n=3 with different cell populations each time as indicated in the figure legend). We felt that a graph would not be representative of our statement that “NCOR1-depleted CRC single cells did not retain the ability to form colonies compared to control cells” in the result section.
If NCOR1-KD produce exactly zero colonies and no fold-reduction can be calculated, then authors can graph the number of colonies and not a fold-reduction number. For this kind of experiment, showing just a representative image is not acceptable. It sounds like colonies from the control cells have not been counted for some reason… Moreover, if authors want to graph a fold-reduction, they can assign to NCOR1-KD cell a figurative value of “1 colony” to be able to make calculation and then explain it in the M&M. Actually, authors can choose whatever ploy they prefer, but for this type of experiment a graph alongside images is mandatory
- These are interesting alternative experiments to be considered in future work.
Thank you for agreeing, but these experiments would have fit perfectly in THIS work, not in another.
- We have included this sentence (lines 402-405): “….in several pathways positively involved in cellular migration and CRC metastasis signaling, including the RHO-GTPase pathway that contribute to cancer progression and metastasis 34 »
This is not a huge deepening of the RNA-seq results. Authors could do better.
- We respectfully disagree with this comment. Even if Caco-2/15 cells express SOX2, it does not mean that this is not a good model to show that absence of NCOR1 does impact on SOX2 expression, which we are in fact demonstrating. A published study, cited in our discussion, has used the parental Caco2 cell line and HT29 cells to demonstrate the involvement of SOX2 in the regulation of cell proliferation (Takeda et al., Sci Rep, 2018).
Takeda et al. studied the role of SOX2 in these cell lines by down-regulating it. Here, authors down-regulate NCOR1 and need to see that SOX2 is up-regulated. These are two different experiments, both considering the final aim and the rationale. Caco-2/15 already express a huge amount of SOX2, which makes improper the use of this cells for this purpose. Citing the work of Takeda for justifying this choice is out of contest.
In addition, SOX2 is commonly up-regulated in both CRC cell lines made deficient in NCOR1, which was an important criteria to focus on this gene among others.
“SOX2 is commonly up-regulated” means that there are other studies (which ones? Why they are not cited in the paper?) demonstrating that SOX2 expression is upregulated after NCOR1-KD? If yes, the novelty of this finding is low…
Additionally, as discussed in our manuscript, SOX2 is an important regulator of cancer stem cells and some evidence point for a role of this regulator in senescence and metastasis potential (Zheng et al., Oncotarget, 2017; Neumann et al., BMC Cancer, 2011). We have made a choice of investigating this target because several groups have performed experiments with SOX2 in CRC cell lines with some overlapping biological effects.
The role of SOX2 in proliferation/metastasis/senescence of CRC cells or cancer generally is not the point. Here the authors claim that SOX2 is the major effector of the NCOR1-KD phenotype only because of its known functions and because its expression is upregulated by NCOR1-KD. This is at least an overstatement. Anyway, if SOX2 is such a fundamental target and a rewiring of the discussion is not acceptable for authors, a deeper functional in vitro evaluation of its relationship with NCOR1 is needed.
Analyzing other targets is limitless given the number of commonly influenced targets in these two cell lines and the rationale of prioritizing any other targets would be similar to what we have decided to do for SOX2.
Functional analysis of a target means that many experiments concerning cells phenotype must be performed, namely proliferation, colony formation, senescence assays etc. Here, to “analyze” SOX2 as a target of NCOR1-KD, authors have only validated that its expression increase after NCOR1-KD. Therefore, such a simple experiment could be performed also for other targets, authors just need other couples of primers and the same RNA samples used for SOX2. Typically, after RNA-seq experiments, qRT-PCR validation of dozens of identified targets is provided.
I deeply respect the authors' opinion and their insight to focus on SOX2, but the goal of a reviewing process is to value the proposed work. Here, many good experiments have been performed and deserve a proper conclusion. A wider validation of RNA-seq targets could be a good way. It would strengthen RNA-seq results and there would be no need for the more complicated in vitro experiments necessary for sustaining a relationship between SOX2 and NCOR1 (as said before). After that, authors could focus the discussion much on SOX2 than on other validated targets, if they wish.
- As stated in the last comment, SOX2 has been already reported to mimic biological effects in CRC cell lines as we have observed (cell proliferation, senescence, metastasis potential, etc.). In our opinion, discussion of additional targets (we have identified more than 400 NCOR1 common targets in both CRC cell lines) could be endless and unfocused in this current manuscript
Refer to the above comment. Just want to point out that 400 common target is a good result. Authors only have to set a Fold Change threshold and validate only the top-scoring genes, about 20 or some more could be a good number.